# Inactivation of Foodborne Pathogens by *Lactiplantibacillus* Strains during Meat Fermentation: Kinetics and Mathematical Modelling

**DOI:** 10.3390/foods12173150

**Published:** 2023-08-22

**Authors:** Seyed Mohammad Bagher Hashemi, Reza Roohi, Masoud Akbari, Alessandra Di Natale, Francesca Conte

**Affiliations:** 1Department of Food Science and Technology, Faculty of Agriculture, Fasa University, Fasa 74681-77375, Iran; 2Department of Mechanical Engineering, Faculty of Engineering, Fasa University, Fasa 74681-77375, Iran; re.roohi@fasau.ac.ir (R.R.); masoud@trn.ui.ac.ir (M.A.); 3Postgraduate School for the “Inspection of Foodstuffs of Animal Origin”, University of Messina, 98122 Messina, Italy; alexdinatale@gmail.com; 4Department of Veterinary Sciences, University of Messina, 98122 Messina, Italy

**Keywords:** fermentation, meat, LAB strains, pathogens, mathematical modelling

## Abstract

This study examined the effect of beef fermentation with *Lactiplantibacillus paraplantarum* (*L*) PTCC 1965, *Lactiplantibacillus* (*L*) *plantarum* subsp. *plantarum* PTCC 1745, and *Lactiplantibacillus* (*L*) *pentosus* PTCC 1872 bacteria on the growth of pathogenic bacteria, including *Salmonella* (S) Typhi PTCC 1609 and *Staphylococcus* (*S*) *aureus* PTCC 1826. The growth of lactic acid bacteria (LAB) and the effect of fermentation on pathogenic bacteria were studied using Weibull: biphasic linear and competitive models. The results showed that the rate of pH reduction was lower in the early stages and increased as the microbial population grew. The *α* parameter was lower for *L. plantarum* subsp. *plantarum* compared to *L. paraplantarum* and *L. pentosus*. The comparison of the α parameter for bacterial growth and pH data showed that the time interval required to initiate the rapid growth phase of the bacteria was much shorter than that for the rapid pH reduction phase. The pH value had a 50% greater effect on the inactivation of *S.* Typhi when compared to the samples containing *L. plantarum* subsp. *plantarum* and *L. pentosus*. The same parameter was reported to be 72% for the inactivation of *St. aureus*. In general, during the fermentation process, LAB strains caused a decrease in pH, and as a result, reduced the growth of pathogens, which improves consumer health and increases the food safety of fermented meat.

## 1. Introduction

Fermentation of meat products results in favorable flavors and enhances the health benefits of these products [1]. In addition, the fermentation process can be used as a very suitable preservation method for these products, which may be superior to some preservation methods for meat products, as many people today are concerned about consuming products containing nitrates because of the possibility of cancer [2,3,4]. Various fermented meat products are manufactured in different countries and regions, including fermented sausage and fermented meat sauce, whose production methods may be regionally specific. Some traditional fermented meat products are sources of probiotics. The literature shows that many studies have focused on fermented meat products containing starter cultures with probiotic activity and potential health benefits [5]. The fermentation process of meat products can be carried out using their natural microbiota or starter cultures, mainly lactic acid bacteria (LAB). These lactic starters include *Pediococcus acidilactici*, *Pediococcus pentosaceus*, lactobacilli, *Enterococcus* strains, and *Leuconostoc* strains [6]. The primary role of these microorganisms is the production of large amounts of lactic acid and small amounts of acetic acid. These bacteria usually do not have strong proteolytic and lipolytic properties [7,8]. Yeasts are also involved in the fermentation of meat products; these yeasts include *Pichia*, *Debaryomyces*, and *Torulopsis* [9]. LAB and yeasts use proteolytic enzymes such as amino acid converting enzymes to create small peptides, free amino acids, and non-protein nitrogen values that are effective in the aroma of fermented meats [10]. Although the rapid production of acid by the starter reduces the microbial risk in fermented meat products, it cannot completely eliminate this concern, especially in products where the fermentation process is slow [11,12]. While the presence of some microorganisms such as aerobic spore formers and *Pseudomonas* strains is not usually a cause for concern, the presence of pathogens such as *Staphylococcus* (*S*) *aureus*, *Escherichia* (*E*) *coli*, and *Salmonella* (*S*) may pose a potential risk to the consumer. For example, in the early stages of the fermentation of meat products, *S. aureus* can produce several enterotoxins and cause food poisoning under the right conditions [13,14]. The outbreaks of verocytotoxigenic *E. coli* have appeared to be associated with fermented meats [15]. *Clostridium botulinum* is one of the dangerous pathogenic microorganisms in meat products that can grow if the pH is above 4.6 and anaerobic conditions are provided. Since the pH is lower than this value in most fermented products, there is no possibility of growth for this microorganism [16].

To better understand the behavior of microorganisms and to compare different microorganisms under different growth conditions, the use of mathematical models in the growth and inactivation of microorganisms can help. By extracting different parameters from the models, it is possible to predict how the microorganisms will behave more accurately. It should be noted, of course, that not all models are suitable for all microorganisms under all conditions, and a model should be chosen that is most appropriate to the behavior of the microorganism [17,18]. A review of the literature shows that only a few studies have been carried out to investigate mathematical models for the fermentation of meat products and the inactivation of pathogens in these products [19,20]. Therefore, this study aims to use *L. paraplantarum* PTCC 1965, *L. pentosus* PTCC 1872 and *L. plantarum* subsp. *plantarum* PTCC 1745 in beef during fermentation and to investigate the inactivation of pathogens including *S.* Typhi PTCC 1609 and *S. aureus* PTCC 1826. In addition, the population of microorganisms and the pH produced by the LAB were modeled by mathematical models such as Weibull, biphasic linear, and competitive models, and the process of the inactivation of pathogens was expressed by the models.

## 2. Materials and Methods

### 2.1. Bacterial Strains

*Lactiplantibacillus paraplantarum* (*L*) PTCC 1965, *Lactiplantibacillus* (*L*) *pentosus* PTCC 1872, and *Lactiplantibacillus* (*L*) *plantarum* subsp. *plantarum* PTCC 1745 were purchased from the collection center of industrial microorganisms (Tehran, Iran) in a lyophilized form and then activated in 20 mL De Man, Rogosa, Sharpe (MRS) broth under anaerobic conditions at 37 °C for 24 h. Pathogenic microorganisms *S.* Typhi PTCC 1609 and *S. aureus* PTCC 1826 were also purchased from the mentioned center and activated in Nutrient Broth (NB) and Trypticase Soy Broth (TSB) for 24 h at 30–37 °C. Then, the activated cultures were centrifuged at 3000× *g* for 10 min at 4 °C (Hettich, UNIVERSAL 320/320 R; Kirchlengern, Germany), and the cells were precipitated individually. The sedimented cells were washed using 0.85% sterile normal saline (NS; 50 mL) solution, and the initial number of colonies was diluted to 10^7^ cfu/mL with 0.85% sterile NS solution.

### 2.2. Fermented Meat Manufacture

Fresh beef was purchased from a store in Shiraz City (Fars Province; Iran) in a sterile form and then cut into pieces (2 cm × 2 cm × 2 cm) under aseptic conditions. In the first stage, parts of the pieces of meat were placed in sterile and closed glass containers, and 2% (*w*/*w*) of salt was added to them. Then, approximately 2% (*v*/*w*) of the LAB strains were added separately to the containers. In the final stage, the remaining pieces of meat were placed in the fermentation vessels, and approximately 2% (*v*/*w*) of both pathogens was added separately. The fermentation process was carried out at 37 °C for 48 h, and the microbial count and pH were measured in meat samples every 8 h. Each sample and all tests were performed three times.

### 2.3. Microbiological Test

For the microbial count, 25 g of each sample was homogenized with 225 mL of 0.1% (*w*/*v*) sterile peptone water using a Stomacher (BagMixer 400W^®^; Interscience, Saint-Nom-la-Bretèche, France). Serial dilutions were made from this homogenate, and further cultures were made in appropriate culture media. LAB strains were cultured on acidified MRS agar with acetic acid. *S.* Typhi and *S. aureus* were grown on MacConkey agar and Baird-Parker agar (egg yolk and tellurite were added and confirmed with the coagulase test), respectively. The incubation was 48 h at 35 ± 2 °C for LAB strains and 24 h at 37 °C for pathogens [21,22].

### 2.4. Determination of pH Value

A portable pH meter (Adwa AD130; Szeged, Romania) was used to measure the pH of the samples.

### 2.5. Mathematical Modelling

Microbial inactivation and population growth, as well as the pH variation trend, were simulated using various mathematical models such as:(1)The Weibull model:

The Weibull model is introduced as an extension of the common linear model. It is used to model the increasing or decreasing trends in microbial variability. This statistical approach has two main parameters: the nominal time scale (*α*) and the shape factor (*β*). The latter determines the rate of variation of the data (the slope of the line in the probability plot) and how the model’s behavior changes over time as the nominal time scale is exceeded. This model was modified to adapt to the current trial:(1)$$log\frac{N}{N_{0}}=-{\left(\frac{t}{\alpha }\right)}^{\beta }$$
where *α* and *β* refer to the nominal time scale and shape factor, respectively [23]. *N*_0_ and *N* (CFU/g) are the population at baseline and after treatment time *t*, respectively.

The biphasic linear model is commonly used in cases where the observed variable has two different behaviors as a function of the influence parameters. It simulates these two trends by dividing the population into two groups with different rates of change [13], with two different subgroups with varying rates of change, as follows [17]:(2)$$log\frac{N}{N_{0}}=log[\Gamma e^{-\varphi_{1}t}+(1-\Gamma )e^{-\varphi_{2}t}]$$
where *Γ*, *φ*_1_ and *φ*_2_ represent the population splitting fraction and variational rate of the subgroups, respectively.

(2)The competitive model:

The variations of *S.* Typhi and *S. aureus* through time were subject to two main factors, namely the growth of the population of these microorganisms and the negative effect of pH. To take into account the competitive impact of these two factors, a mathematical model has to be implemented using the semi-empirical parameters obtained from the experimental measurements. The Jameson effect [18] and the Huang model [19] were used for population growth in the presence of LAB strains, while a direct reducing trend was included for the pH. For this, the population growth rate of *S.* Typhi and *S. aureus* was determined based on the logistic model [24] using the ordinary differential equations (ODEs) as:(3)$$\frac{d[X]}{dt}=\mu_{max}\left(1-\frac{\left[X\right]}{{\left[X\right]}_{max}}\right)\left[X\right]-k_{PH}(7.0-PH)$$
where $\frac{d[X]}{dt}$ is the rate of bacterial growth, [*X*] is the bacterial concentration at time *t*, $\mu_{max}$ is the maximum specific growth rate of the microorganism [1/h], ${\left[X\right]}_{max}$ is the maximum population count (in log CFU/g) and $k_{PH}$ is the rate of decomposition due to pH reduction.

Thus, according to the model described, the effects of the initial population growth of *S.* Typhi and *S. aureus* and their subsequent decline due to pH reduction are considered in this model. It should be noted that Vrancken et al. [24] only used the first term on the right side of Equation (3), and the second term is added in the present modelling to consider the effect of pH as a linear correlation on the population change rate.

### 2.6. Mathematical Considerations

To assess the accuracy of the implemented models, the adjusted *R*^2^(*Adj-R*^2^) and the root mean square error (*RMSE*) measures are implemented. The *Adj-R*^2^ criterion is formulated as follows:(4)$$AdjustedR^{2}=1-[\frac{\left(m-j)(1-R^{2}\right)}{m-o}]$$
where *m* is the number of measured data, *o* is the number of the parameter, and *j* is the indicator variable with values 1 and 0, respectively, in the case of an intercept and other cases [25].

Furthermore, the difference between the observation and prediction data is measured by the *RMSE* as follows:(5)$$RMSE=\sqrt{\frac{{\left(observed-predicted\right)}^{2}}{n-p}}$$

For statistical analysis, ANOVA and Duncan’s multiple range test were performed. The *p*-value measure was used to determine the statistical significance of the coefficients in the regression models. For all models, *p* < 0.05 was considered statistically significant. For the competition model, the governing equation was solved using the 4th order Runge-Kutta using MATLAB’s (version: 2013b v8.2) ode45 solver. To determine the unknown parameters, namely, $\mu_{max}$, ${\left[X\right]}_{max}$ and $k_{PH}$, a nonlinear least-square optimization algorithm is implemented. Nonlinear Least Squares, as an optimization technique, is used to develop regression models related to data sets, including nonlinear features. Models for such data sets are nonlinear in their coefficients. The parameters obtained were then adjusted iteratively to reduce the deviation of the numerical results obtained from the experimental data.

## 3. Results and Discussion

The population growth of the lactobacilli strains is shown in Figure 1. The population growth rate was lower in the first 9 h, followed by a linear increase with time up to about 32 h and a lower growth rate in the final phase after 32 h. The highest and lowest population increases were for *L. plantarum* subsp. *plantarum* and *L. pentosus*, respectively. It was observed that the logarithmic population of *L. paraplantarum*, *L. plantarum* subsp. *Plantarum*, and *L. pentosus* were increased by 3.5, 3.5, and 2.9 log, respectively, after 48 h.

The growing lactobacilli strains caused the pH to drop (Figure 2). The rate of pH reduction was lower in the early stages and tended to increase as the microbial population increased. The highest and lowest pH decreases occurred in the presence of *L. plantarum* subsp. *plantarum* and *L. pentosus*, respectively. Decreased pHs of approximately 1.5, 2.1, and 2.4 were observed due to the population growth of *L. pentosus*, *L. paraplantarum*, and *L. plantarum* subsp. *plantarum*, respectively.

To examine the effect of the microbial population on the pH, the data obtained are shown in Figure 3.

According to the results obtained, the dependence of the pH on the microbial population was almost the same for all the bacteria studied. However, the pH reduction effect was slightly stronger for *L. plantarum* subsp. *plantarum* than for *L. paraplantarum* and *L. pentosus*. For example, at a population of 8 log CFU/g, the pH reduction was 0.5, 0.7, and 0.9 for *L. pentosus*, *L. paraplantarum*, and *L. plantarum* subsp. *plantarum*, respectively. However, for logarithmic populations of below 7 and above 9, all microorganisms had an identical effect on pH (as the plots coincide at these intervals).

The effect of *L. paraplantarum*, *L. plantarum* subsp. *Plantarum*, and *L. pentosus* microorganisms on the population of *S*. Typhi and *S. aureus* bacteria is shown in Figure 4a–c. As can be observed, two consecutive phases can be observed in all the cases studied. Initially the population of *S.* Typhi and *S. aureus* increased at a lower rate. In the second phase, by increasing the competitive factors, namely the population of lactobacilli strains and the lowering pH, a steep reduction trend occurred. Specifically, *S. aureus* was more sensitive to the competitive factors, and its population growth trend stopped and reversed earlier compared to *S.* Typhi.

The population of both pathogens were decreased after about 30 h of incubation time. The population decrease beyond that instance is between 4.3 and 5 log for *S.* Typhi and 5.1 and 5.5 log for *S. aureus*.

The parameters of the Weibull model for the variation of *L. paraplantarum*, *L. plantarum* subsp. *Plantarum*, and *L. pentosus* microorganisms and pH are presented in Table 1 and Table 2.

Regarding the growth of bacterial numbers, it can be observed that the *α* parameter (which indicates the nominal time scale) was lower for *L. plantarum* subsp. *plantarum* compared to *L. paraplantarum* and *L. pentosus*. As *α* indicates the onset of the rapid population growth, it can be concluded that *L. plantarum* subsp. *plantarum* is the first bacterium to initiate the main growth phase, followed by *L. paraplantarum* and *L. pentosus*. It should also be noted that parameter *β* is almost the same for all the lactobacilli strains studied. Regarding the size of the statistical evaluation parameters (i.e., *Adj-R*^2^ and *RMSE*), the Weibull model is suitable for simulating the measured data.

Like the population growth, the same observation can be made for pH, except that the nominal time scale is different. The comparison of the *α* parameter for population growth and pH data showed that the time interval required to initiate the rapid growth phase of the bacteria was considerably less than that for the rapid pH decrease phase. For example, the *α* parameter for population growth *L. plantarum* subsp. *plantarum* was 12.74, which was significantly lower than the 25.61 obtained for the pH variation.

The parameters of the biphasic model for the variation of pH as a function of microbial population (namely, *L. paraplantarum*, *L. plantarum* subsp. *Plantarum*, and *L. pentosus*) are presented in Table 3. It is evident that the biphasic model can be implemented to model pH as a function of the microbial population with sufficient accuracy.

The parameters of the competitive model are listed in Table 4 and Table 5 for *S.* Typhi and *S. aureus*, respectively.

The physical interpretation of the listed parameters should be considered to extract the appropriate information from the calculated data. $k_{PH}$ indicates the importance of pH for the inactivation of *S.* Typhi and *S. aureus*.

According to the results, the magnitude of this parameter was higher for *L. pentosus* compared to *L. paraplantarum* and *L. plantarum* subsp. *plantarum* for the inactivation of both *S.* Typhi and *S. aureus*. Specifically, pH had a 50% greater effect on the *S.* Typhi inactivation than the samples containing *L. plantarum* subsp. *plantarum* and *L. pentosus*. The same parameter was reported to be 72% for *S. aureus* inactivation.

The $\mu_{max}$ and ${\left[X\right]}_{max}$ parameters had a significant influence on the first phase of *S.* Typhi and *S. aureus* population variation (i.e., growth phase), V. A higher $\mu_{max}$ indicates increased variation rates in the first phase, while a higher ${\left[X\right]}_{max}$ is a measure of the overpopulation limit. As ${\left[X\right]}_{max}$ increased, the limit beyond which the negative effect of the population on microbial growth rate occurs was raised.

For the inactivation of *S.* Typhi, the presence of *L. paraplantarum* causes the highest growth rate in the first phase, while the overpopulation limit was the highest for the sample containing *L. pentosus*. The maximum growth rate and overpopulation limit for *S. aureus* belong to the samples containing *L. pentosus* and *L. plantarum* subsp. *plantarum*, respectively.

The logarithmic population of *L. paraplantarum*, *L. plantarum* subsp. *Plantarum*, and *L. pentosus* were increased by 57.3, 56.4, and 46.7%, respectively, after 48 h. Based on the results obtained, a reduction in the pH of 1.5, 2.1, and 2.4 was observed due to the population growth of *L. pentosus*, *L. paraplantarum*, and *L. plantarum* subsp. *plantarum*, respectively.

During the fermentation process, LAB strains can create unfavorable conditions for the growth of pathogenic microorganisms, thus preventing their growth. LAB strains affect the growth of pathogens by producing lactic acid and lowering the pH, bacteriocin, and hydrogen peroxide. Of course, not all LAB strains produce all these compounds; for example, they may only produce lactic acid and affect pathogen growth [26,27,28]. In this research, it was observed that the production of acid decreased the pH of the growth environment of LAB strains, and as a result, it had a negative effect on the growth of pathogens. Other inhibitory factors produced by LAB strains were not investigated in this research. On the other hand, the growth of pathogens in fermented products, such as meat, can be affected by factors such as salt, moisture content, and inhibitors such as nitrite, in addition to starters [29], which were of course kept constant for all samples in this study.

The literature indicates that acid production is one of the main factors influencing the inactivation of pathogens by LAB in fermented meat and meat products. In some cases, mathematical models have been used in the studies. For example, Tabanelli et al. (2016) [30] reported that pH decreases in Italian fermented sausages due to producing acids such as lactic and acetic acid by starter LAB [30]. The inactivation of *Salmonella* spp. in dry fermented sausages was studied using a Weibull model and a polynomial equation. The authors observed that the acid production by the starter affected the time to first log reduction (*δ*) but did not affect the overall shape (*p* parameter) of the inactivation [20]. Quinto et al. (2016) [19] investigated the effect of co-culturing *Latilactobacillus sakei* with *Listeria* (*L*) *monocytogenes* in a model of meat gravy. The Verhulst logistic model was modified; the model showed that the co-culture of *L. monocytogenes* with *L. sakei* in different inoculations and temperatures reduced the growth of *L. monocytogenes*. The simple logistic model also showed that the effect of *L. sakei* on *L. monocytogenes* was independent of environmental conditions [19]. Huang et al. (2022) [31] investigated the competition between lactic acid bacteria and *L. monocytogenes* during the simultaneous fermentation and drying of meat sausages. The results showed that *L. plantarum* and *Levilactobacillus brevis* strains could suppress the growth of *L. monocytogenes*. The interaction between LAB strains and *L. monocytogenes* was described by a modified Lotka–Volterra equation [31]. In another study, *L. sakei* CTC 494 and *Latilactobacillus curvatus* LTH 1174 reduced *Lactobacillus innocua* LMG 13,568 by more than 3 log CFU/mL during two days of fermentation, which was attributed to the production of bacteriocin by these microorganisms [32]. The effect of *Enterococcus* (*E*) *mundtii* CRL35 on *L. monocytogenes* in meat fermentation was studied in a meat model system. It was observed that *E. mundtii* can be used as a bioprotective agent in fermented sausage by producing acid and bacteriocin [33]. Pragalaki et al. (2013) [34] investigated the inactivation of *L. monocytogenes* and *E. coli* O157:H7 by *L. sakei* strains during sausage fermentation. The results showed that the inactivation of *L. monocytogenes* by *L. sakei* strains was significant compared to the control samples, and a 2.2 log reduction was observed for *E. coli* O157:H7. Bacterial inactivation was modeled using a “shoulder plus log-linear regression” [34].

## 4. Conclusions

In this study, lactobacilli strains were used for beef fermentation. Their inhibitory effect on the growth of pathogenic bacteria, including *S.* Typhi and *S. aureus*, was investigated using mathematical models, including Weibull, biphasic linear, and competitive models. The pH reduction effect was slightly stronger for *L. plantarum* subsp. *plantarum* compared to *L. paraplantarum* and *L. pentosus*. *S. aureus* was more sensitive to competitive factors, and its population growth trend stopped and reversed earlier than the growth trend of *S.* Typhi. For the inactivation of *S.* Typhi, the presence of *L. paraplantarum* caused the highest growth rate in the first phase, while the overpopulation limit was the highest for the sample containing *L. pentosus*. The maximum growth rate and overpopulation limit for *S. aureus* belong to the samples containing *L. pentosus* and *L. plantarum* subsp. *plantarum* microorganisms, respectively.

The products mentioned as examples may pose a potential risk to the consumer. Therefore, preservation by inoculating an appropriate type and number of LAB cultures is paramount to protecting product safety and consumer health. Further studies using mathematical modelling are needed to investigate the effect of different parameters in preventing pathogen growth during the fermentation process, especially in short-ripened fermented meat products.

## Figures and Tables

**Figure 1 foods-12-03150-f001:**
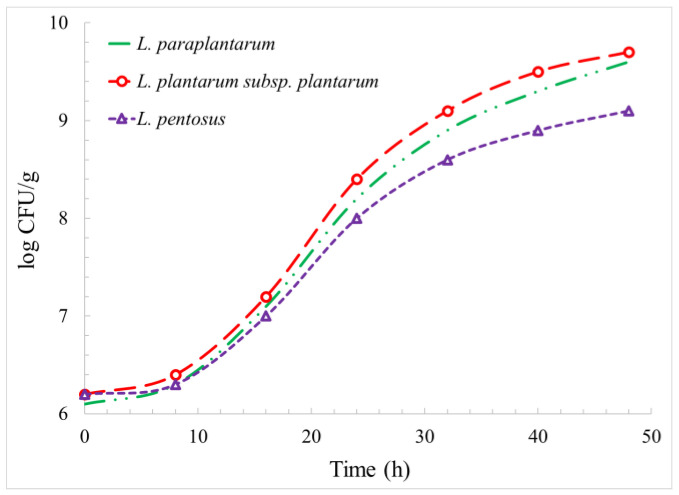
Population growth of *L. paraplantarum*, *L. plantarum.* subsp. *Plantarum*, and *L. pentosus*.

**Figure 2 foods-12-03150-f002:**
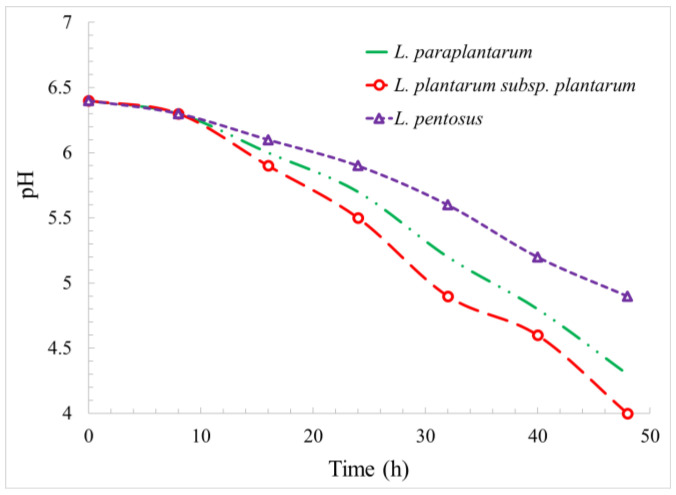
Reduction of pH in the presence of *L. paraplantarum*, *L. plantarum* subsp. *Plantarum*, and *L. pentosus* as a function of time.

**Figure 3 foods-12-03150-f003:**
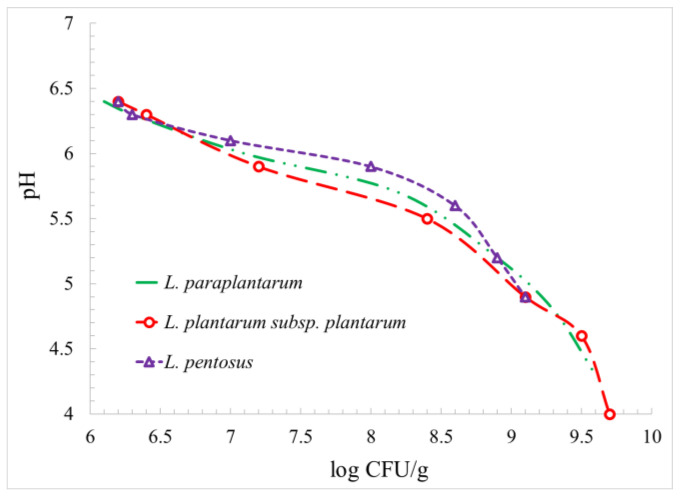
Reduction in pH in the presence of *L. paraplantarum*, *L. plantarum* subsp. *Plantarum*, and *L. pentosus* as a function of log population.

**Figure 4 foods-12-03150-f004:**
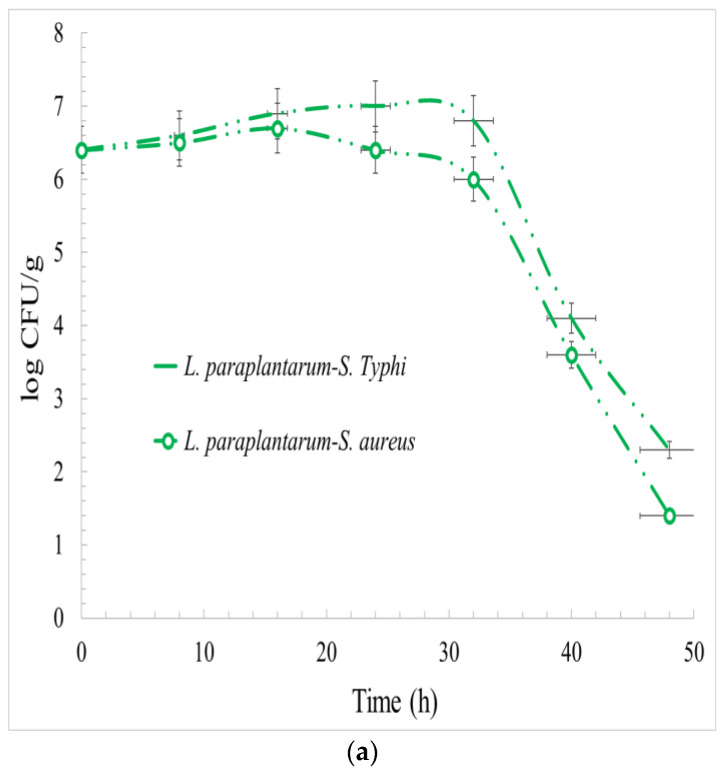
Effect of the presence of *L. paraplantarum* (**a**), *L. plantarum* subsp. *plantarum* (**b**), and *L. pentosus* (**c**) on the population of *S.* Typhi and *S. aureus*.

**Table 1 foods-12-03150-t001:** Parameters of the Weibull model for the variation of *L. paraplantarum*, *L. plantarum* subsp. *Plantarum*, and *L. pentosus* microorganisms.

Model Parameters	Bacterial Species		
	*L. paraplantarum*	*L. plantarum* subsp. *plantarum*	*L. pentosus*
*α*	13.3 ^b^	12.74 ^c^	15.82 ^a^
*β*	1.034 ^a^	1.012 ^a^	1.039 ^a^
**Statistical evaluation parameters**	**Values**		
*Adj-R* ^2^	0.9567	0.9452	0.9404
*RMSE*	0.2988	0.3434	0.2989

All values are means of three determinations with coefficient of variations (CV = SD/mean × 100) <5%. For each model parameter, means with the same lowercase letters are not significantly different at *p* < 0.05.

**Table 2 foods-12-03150-t002:** Parameters of the Weibull model for pH variation.

Model Parameters	Bacterial Species		
	*L. paraplantarum*	*L. plantarum* subsp. *plantarum*	*L. pentosus*
*α*	29.35 ^b^	25.61 ^c^	36.49 ^a^
*β*	1.523 ^a^	1.4 ^a^	1.53 ^a^
**Statistical evaluation parameters**	**Values**		
*Adj-R* ^2^	0.9979	0.9921	0.9974
*RMSE*	0.03609	0.08032	0.02893

All values are means of three determinations with coefficient of variations (CV = SD/mean × 100) <5%. For each model parameter, means with the same lowercase letters are not significantly different at *p* < 0.05.

**Table 3 foods-12-03150-t003:** The parameters of the biphasic model for pH variation as a function of *L. paraplantarum*, *L. plantarum* subsp. *Plantarum*, and *L. pentosus* population.

Model Parameters	Bacterial Species		
	*L. paraplantarum*	*L. plantarum* subsp. *plantarum*	*L. pentosus*
*Γ*	0.997 ^a^	0.996 ^a^	0.912 ^b^
*φ* _1_	0.31 ^b^	0.4452 ^a^	0.2746 ^c^
*φ* _2_	−0.881 ^c^	−2.381 ^c^	−1.943 ^b^
**Statistical evaluation parameters**	**Values**		
*Adj-R* ^2^	0.9955	0.9924	0.9856
*RMSE*	0.05314	0.07912	0.06768

All values are means of three determinations with coefficient of variations (CV = SD/mean × 100) <5%. For each model parameter, means with the same lowercase letters are not significantly different at *p* < 0.05.

**Table 4 foods-12-03150-t004:** Parameters of the *S.* Typhi in the presence of *L. paraplantarum*, *L. plantarum* subsp. *Plantarum*, and *L. pentosus*.

Statistical Parameters	*S.* Typhi		
	*L. paraplantarum*	*L. plantarum* subsp. *plantarum*	*L. pentosus*
$\mu_{max}$	0.0180 ^a^	0.0140 ^b^	0.0136 ^c^
${\left[X\right]}_{max}$	11.3600 ^c^	13.9000 ^b^	15.7600 ^a^
$k_{PH}$	0.0046 ^b^	0.0040 ^b^	0.0060 ^a^

All values are means of three determinations with coefficient of variations (CV = SD/mean × 100) <5%. For each model parameter, means with the same lowercase letters are not significantly different at *p* < 0.05.

**Table 5 foods-12-03150-t005:** The competitive model parameter for *S. aureus* in the presence of *L. paraplantarum*, *L. plantarum* subsp. *Plantarum*, and *L. pentosus* microorganisms.

Statistical Parameters	*S. aureus*		
	*L. paraplantarum*	*L. plantarum* subsp. *plantarum*	*L. pentosus*
$\mu_{max}$	0.0100 ^a^	0.0056 ^b^	0.0106 ^a^
${\left[X\right]}_{max}$	11.1200 ^c^	13.7000 ^a^	12.6000 ^b^
$k_{PH}$	0.0044 ^b^	0.0036 ^c^	0.0062 ^a^

All values are means of three determinations with coefficient of variations (CV = SD/mean × 100) <5%. For each model parameter, means with the same lowercase letters are not significantly different at *p* < 0.05.

## Data Availability

Data are available in the paper.

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
