# Peer review of "Inactivation of Foodborne Pathogens by Lactiplantibacillus Strains during Meat Fermentation: Kinetics and Mathematical Modelling"

_foods, 2023, doi:10.3390/foods12173150_

Round 1
Reviewer 1 Report
The manuscript titled “Inactivation of foodborne pathogens by Lactobacillus strains during meat fermentation: Kinetics and Mathematical Modelling” investigated the impact of Lactobacillus strains on the growth of foodborne pathogens during meat fermentation. L. paraplantarum, L. plantarum subsp. plantarum, and L. pentosus were studied for their inhibitory effects on Salmonella Typhi and Staphylococcus aureus. Mathematical modeling was employed to analyze pH reduction and pathogen inactivation. Results showed a correlation between microbial growth, pH reduction, and pathogen inactivation. L. plantarum subsp. plantarum exhibited quicker growth initiation, while pH reduction had a substantial impact on pathogen inactivation. The study highlights the potential of LAB strains for enhancing food safety in meat fermentation processes. Though the study has significance in the area, but massive improvement is required as follows:
abstract
1. briefly highlight the novel aspects or contributions of your study in terms of food safety and consumer health, particularly regarding the inhibition of foodborne pathogens during meat fermentation.
2. provide a bit more detail about the specific findings or trends observed in terms of the reduction of pathogen populations and the corresponding changes in pH during the fermentation process.
3. In what ways does your study's approach to using mathematical modeling enhance the understanding of the interactions between LAB strains and pathogenic bacteria during meat fermentation?
Introduction:
1. Are there specific sources provided to validate claims about health benefits and concerns regarding nitrates in fermented products?
2. Could more concise explanations about the roles and importance of LAB and yeasts in the fermentation process be incorporated?
3. How can the explanation of risks posed by pathogens like Staphylococcus aureus and Escherichia coli be further elaborated for better reader understanding?
4. What additional context can be provided to better underscore the importance of mathematical modeling and its relevance to understanding microbial behavior?
5. How can the objectives of the study be clearly outlined, specifying the purpose of using specific microorganisms and investigating pathogen inactivation?
Methodology Sections: please response the questions in the relevant sections to improve the methods:
1. The description of bacterial strains' activation could be more concise and straightforward.
2. How can the steps involved in preparing fermented meat be organized more systematically to ensure a clear and easy-to-follow description?
3. The part explaining model assessment and parameters needs to be more concise and explicit.
4. How can complex terms like "nonlinear least-square optimization algorithm" be broken down or supplemented with brief explanations to ensure reader understanding?
5. How can the flow between subsections (e.g., bacterial strains, preparation of fermented meat, microbiological assay) be improved for a more logical progression?
Results and discussion:
1. please expand your discussion by comparing results with findings from similar research in the literature, highlighting both consistencies and differences to strengthen the validity of your conclusions.
2. Please provide deeper mechanistic insights into how the Lactobacillus strains influence pathogen growth, particularly focusing on the specific compounds they produce and their interactions with factors such as pH and other inhibitors?
3. please address potential variability in your results by discussing any limitations in your experimental setup or methodology that might have affected the observed trends in population growth, pH reduction, and pathogen inactivation?
Conclusion:
1. Conclusion should be concise, strengthen the linkages between objectives and results and highlighting insights gained.
english language should be improved
Author Response
REVIEWER 1
All recommendations and specific comments were taken into account and the paper was modified accordingly. We have modified the text and added the references. Changes are marked in red colour.
We are grateful for your thoughtful suggestions who helped shape this paper.
KInd regards,
Authors

Reviewer 2 Report
The authors present an interesting piece of work, that is more of a short communication than a full research paper, as the body of research reported is limited. The manuscript still needs a lot of work, as the materials and methods section does not give important information to evaluate the validity of the conclusions the authors derive from their data. For instance, it is not stated anywhere in the manuscript how many replicates of each experiment were made. Also, the authors fail to acknowledge the strong limitations of the methodologies they chose to use for the bacterial counts. McConkey agar is not the best medium to count salmonellae and this could have led to results that are not what the authors imply them to be. The use of a non-buffered diluent might also explain why no differences were observed among the lactic acid bacterial strains used regarding pathogen counts in meat, and why did both the staph and the salmonella – two very different bacteria regarding their resistance to environmental factors – performed similarly in the challenge tests. Therefore, the authors need to completely overhaul their manuscript. Furthermore, the authors did not use the Foods template for manuscripts and did not insert line numbers, making it harder to review the paper. In further review rounds, please comply with the instructions for authors of the journal.
Page 3, lines 2 – 5: Please rewrite. Avoid unnecessary repetition of words within the same sentence (e.g., such as the use of nitrates … products containing nitrates.
Page 3, line 11: Please remove the unnecessary stop, in red, before [4].
Page 3, line 12: Please replace the term “flora” with “microbiota”. Flora is an atavism, from the time microbes were thought to belong to the Plant Kingdom. For the language of the manuscript to be precise, the term “microbiota” should be used instead.
Page 3, line 14: “Lactobacillus” – please replace with “lactobacilli”. The authors cited wrote their paper at a time when the genus Lactobacillus comprised strains that are nowadays placed in other genera (such as the Lactiplantibacillus mentioned earlier in this manuscript). To stress the idea that the authors do not mean to mention only the strains within species that are part of the present Lactobacillus genus, it is best to use the non-taxonomic name of lactobacilli, which would encompass also the strains that are presently placed in the genera established by Zheng et al. in 2020.
Page 3, line 14: Please correct Leuconastoc to Leuconostoc.
Page 3, lines 14 – 16: Lactic and acetic acids do not have a large impact on flavour; they mainly influence the perception of acidity by the consumers. Lactic acid bacteria do produce, however, other compounds with low perception thresholds and a large impact in fermented meat products. The authors should revise this sentence, discussing better the relationship between fermented meat microbiota and flavour of the resulting products.
Page 3, lines 17 – 18: Please explain, concisely, the importance of proteolysis and lipolysis in meat fermentation.
Pages 3 (lines 22 – 23) and 4 (lines 1 – 3): Toxinogenic clostridia, such a Clostridium botulinum, are an important microbial contaminant that threatens the safety of sausages. The authors should briefly discuss it in their introduction text, explaining why they decided that it is not important, in the context of the research they present in this manuscript, to test their lactic acid bacteria for antimicrobial activity against this important foodborne pathogen.
Page 4, lines 9 – 11: The authors state that “A review of the literature shows that only a few studies have been carried out …”. However, no citations/references are given to substantiate this claim. Please insert the relevant citations/references. You could provide, for instance, the citation/reference to a couple of very recent review papers on this subject or, alternatively, insert the few references found on mathematical models applied to meat fermentations.
Page 4, item 2.1, line 5: Please be consistent with the use of abbreviations. Formerly, you used St. as an abbreviated form of the word Staphylococcus. In this line, a different abbreviation is used. Please stick to one abbreviation throughout the text.
Page 5 – line 1: “3000 x g”; the symbol used is the letter x. Please use the correct multiplication symbol (×).
Page 5, lines 1 – 4: What was the final cell density of the bacterial suspensions obtained? This is an important datum, and should be given in the text, either as a McFarland value or as cfu ml-1 of suspension.
Page 5, item 2.2. – What was the initial cell density obtained for the lactic acid bacteria and for the pathogens (as cfu ml-1)? This is a very important piece of data that the authors should include. It is essential for reviewers – and latter on, for readers – to evaluate whether this challenge test was properly planned and also to help formulating conclusions from the obtained data.
Equally important – how many repetitions were prepared for each type of sample?
Were both pathogens (staph and salmonella) added together or separately?
This part of the manuscript lacks a lot of detail.
Page 5, item 2.3. – I have serious concerns about the counting methodologies used. First of all, lactic acid bacteria were added to the samples, which caused a change in their pH. Therefore, the diluent used should have had a good buffering capacity, to avoid damage to the bacterial cells during the sample preparation procedure, with consequences upon the counts obtained, which would misrepresent the death of the studied pathogens. Another concern regards the use of McConkey Agar for counting salmonellae. This is not a good medium for this purpose, as several other non-lactose fermenting microorganisms abound in meats and would also distort the obtained counts. Finally, which of the different formulations of Baird-Parker Agar was used? Was it the formulation that has added rabbit plasma? If not, if it had only egg yolk and tellurite added, confirmation of the coagulase-positive character of the obtained typical colonies is necessary. The authors simply do not give detail enough on this aspect.
Page 8, item 3. (Results and discussion), line 1: Please revise the use of the word “Lactobacillus” in this case, for the reasons explained above.
Page 8, item 3, lines 4 – 6: When using logarithmic counts, it makes more sense to mention the number of log cycles than percentages. Also, how many replicates were prepared per sample type? This information is extremely important for the interpretation of the graph in figure 1. Not much difference can be seen between both L. paraplantarum strains used, but the L. pentosus reached final populations in the meat that were considerably lower (ca. 1 log cycle). Were these differences statistically significant? Why was not an ANOVA performed to assess the significance of the differences in these counts? Were there enough repetitions for the ANOVA to be possible?
Page 8, text on figure 2 (pH values): in accordance with the bacterial counts, the pH drop is deeper in the L. paraplantarum strains than in the L. pentosus. The same questions apply here as with the bacterial counts: why was not an ANOVA performed?
Page 10, last line: please reconsider the use of the word “Lactobacillus” in this context. I would recommend using “lactobacilli” instead, for the reasons explained above.
Last paragraph of page 10 – first paragraph of page 11: The authors fail to acknowledge the main findings related to their figure 3, which are that the populations of both pathogens tested decreased only after 30 h of incubation time, and that the decrease was, in all cases, close to 5 log, an important decrease. Also, they should have discussed the fact that, in spite of its lower growth and less deep decrease in pH, L. pentosus achieved a deep decrease in the number of pathogens, indicating that other antimicrobial factors could have been at play (e.g., bacteriocin production). The authors need to acknowledge here the potential flaws in their experimental design, namely the use of a medium (MRS) that is not selective, but rather elective, for lactic acid bacteria*, the use of a non-optimal medium for salmonella enumeration**, and, if applicable, the shortcomings of the Baird-Parker formulation used. Also, the use of a non-buffered diluent might account, at least partially, for such deep decreases in pathogen populations, which are, surprisingly, similar both in salmonella and in staph.
* Using Rogosa Agar, if your lactic acid bacterial strains grow well in it, would have provided a bit more of selectivity, and adding a suitable pH indicator could have aided in differentiating the lactic acid bacteria from the many others (including the tested pathogens) that grow in this medium.
** This can be done by changing the text and the graph legends to “McConkey counts” instead of “S. Typhi”.
Throughout page 16 – Please revise the use of the word “Lactobacillus”, as explained above.
Page 16, line 17 – Lactobacillus sakei is now called Latilactobacillus sakei. Please correct the text accordingly. Also, please italicize the genus name in this species.
Page 17, line 2 – Since it is the first time you mention the species L. brevis, please give its full name (Levilactobacillus brevis).
Page 17, line 3 – Please revise the use of the word Lactobacillus in this context, according to what was explained above.
Page 17, line 4 – Please correct the name of L. curvatus. Also, since this is the first time you mention this species, please give its full name (Latilactobacillus curvatus). Equally, please gie the full name of L. innocua when you mention this species for the first time.
Page 17, item 4 (Conclusions): Again, revise the use of the word Lactobacillus in this context. Most of the text presented under conclusions belongs in the discussion section. Conclusions need to be short, concise, and refer almost exclusively to the results obtained by the authors, which is not the case.
Some sentences are wordy, and some need revising to improve their structure.
Author Response
REVIEWER 2
All recommendations and specific comments were taken into account and the paper was modified accordingly. We have modified the text and added the references. Changes are marked in red colour.
We are grateful for your suggestions who helped shape this paper.
Kind regards,
Authors

Round 2
Reviewer 1 Report
the authors revised well, therefore it can be accepted for publications
Minor english edition is required